# In Silico Prediction of Skin Permeability Using a Two-QSAR Approach

**DOI:** 10.3390/pharmaceutics14050961

**Published:** 2022-04-28

**Authors:** Yu-Wen Wu, Giang Huong Ta, Yi-Chieh Lung, Ching-Feng Weng, Max K. Leong

**Affiliations:** 1Department of Chemistry, National Dong Hwa University, Shoufeng, Hualien 974301, Taiwan; 410512071@gms.ndhu.edu.tw (Y.-W.W.); 810812203@gms.ndhu.edu.tw (G.H.T.); 410712015@gms.ndhu.edu.tw (Y.-C.L.); 2Institute of Respiratory Disease and Functional Physiology Section, Department of Basic Medical Science, Xiamen Medical College, Xiamen 361023, China; cfweng-cfweng@hotmail.com

**Keywords:** in silico, permeability constant, hierarchical support vector regression (HSVR), partial least square (PLS)

## Abstract

Topical and transdermal drug delivery is an effective, safe, and preferred route of drug administration. As such, skin permeability is one of the critical parameters that should be taken into consideration in the process of drug discovery and development. The ex vivo human skin model is considered as the best surrogate to evaluate in vivo skin permeability. This investigation adopted a novel two-QSAR scheme by collectively incorporating machine learning-based hierarchical support vector regression (HSVR) and classical partial least square (PLS) to predict the skin permeability coefficient and to uncover the intrinsic permeation mechanism, respectively, based on ex vivo excised human skin permeability data compiled from the literature. The derived HSVR model functioned better than PLS as represented by the predictive performance in the training set, test set, and outlier set in addition to various statistical estimations. HSVR also delivered consistent performance upon the application of a mock test, which purposely mimicked the real challenges. PLS, contrarily, uncovered the interpretable relevance between selected descriptors and skin permeability. Thus, the synergy between interpretable PLS and predictive HSVR models can be of great use for facilitating drug discovery and development by predicting skin permeability.

## 1. Introduction

The skin, which is the largest organ in human body [1] due to the fact that it has the largest surface and accounts ca. 15% of adult body weight [2], provides a major barrier against the external environment from the internal environment [3]. It is composed of multiple layers, namely the surface epidermis, the deeper dermis, and the innermost subcutis [4], which, in turn, have different constructs, morphology forms, and functions [5]. The hydrophobic stratum corneum (SC), which is the uppermost layer of epidermis, plays a predominant role in barrier to skin permeation and is normally regarded as the “rate-limiting step of permeation” [6].

Topical and transdermal drug delivery only accounts a small portion of administration routes [7]. Nevertheless, it has become an attractive and preferred route of therapeutic delivery partly due to its noninvasive nature and more desirable safety profiles [8,9]. For instance, it has been proposed to use patches to deliver insulin [10] and the pandemics COVID-19 vaccine [11]. Furthermore, it can provide extra clinical benefits as compared with the other administration routes. For instance, it is not uncommon to observe postoperative nausea and vomiting (PONV) after general or regional anesthesia [12] and it is normal to treat with scopolamine (hyoscine) [13], which is associated with various undesirable side-effects, such as xerostomia, blurriness, drowsiness, vertigo, or hallucinations in some cases [14]. Those anticholinergic symptoms, nevertheless, can be avoided in case of administration by transdermal patch [15]. In fact, scopolamine was the first marketed transdermal patch [16].

Additional clinical benefits offered by topic administration can be illustrated by the fact that topic administration can be totally exonerated from the potential adverse side-effects associated with the first pass effect (FPE) in the liver when administrated orally as well as the variations in gastrointestinal (GI) tracks, namely pH discrepancies, food intake, stomach emptiness [17].

Skin permeability is a pivotal factor that should be taken into account in the pharmaceutical and cosmetics industries for optimization of the delivery of active substances as well as hazard and risk evaluation [18]. Various in vitro, in vivo, and ex vivo assay systems have been devised to assess drug retention in skin layers and skin permeability [19]. Of various in vitro assay systems, skin from human, pig, hairless rodent, guinea pig, and artificial membrane are acceptable by the European Medicines Agency (EMA) as a means to evaluate the skin permeability [20]. Nevertheless, ex vivo excised human skin is still considered as the de facto standard for in vitro permeation assessments despite the fact that there are a number of ethical issues associated with it [21]. In vitro skin permeability is normally defined by the permeability coefficient or constant (*K*_p_) as follows,
(1)$$K_{p}=\frac{J_{\mathrm{ss}}}{\Delta C_{v}}$$
where *J*_ss_ and Δ*C*_V_ are the steady state flux (*J*_ss_) and the chemical concentration difference (Δ*C*_V_), respectively [22].

In silico modeling provides an interesting alternative to assess skin permeability since it is less time-consuming and economically efficient in addition to the fact that there are no ethical issues when compared with its in vivo and in vitro counterparts [23]. Most importantly, in silico technology can be applied to the virtual compounds, *viz*. compounds that have not been synthesized yet. In fact, numerous in silico models, to predict skin permeability, have been published [24,25,26,27,28,29,30,31,32,33,34,35,36,37,38].

Skin permeation can take place through the transcellular route, in which the permeants cross SC, the intercellular route, in which the permeants across the lipid matrix, and the shunt or appendageal route, in which the diffusion goes into the hair follicles, sebaceous gland, and sweat gland [20] as illustrated by Figure 1 of Benson [39]. Furthermore, compounds with different physicochemical properties can penetrate skin layers via different routes. For instance, very polar, mediate polar, and poor polar compounds can exhibit different permeation behavior and a sophisticated theoretical model that can take into account the diverse mechanisms accommodated by solutes of different polarities is needed as suggested [24]. The ATP-binding cassette (ABC) superfamily and solute carrier (SLC) superfamily can be expressed in human skin [40] that can further enhance and/or reduce permeability, making the production of a sound in silico model that can take into account all those complicated factors extremely difficult, if not impossible. Theoretical models based on in vitro assays, except for human skin, can be of limited applicability due to poor or little correlation between human skin and skin of the other animal species.

To date, most of the quantitative in silico models are constructed using either of two categories, namely the linear regression or machine learning (ML) schemes [41]. The former including partial least square (PLS) and multiple linear regression (MLR) can render the link between adopted descriptor and biological activity [41]. Nevertheless, it is hard for linear models to properly function when such links are very complicated as exemplified by the varied weights between molecular polarity and skin permeability (*vide supra*). This difficulty can be appropriately addressed by ML-based schemes since ML-based models generally perform better than their linear counterparts in handling nonlinearity [42]. This “black box” approach, conversely, makes ML models difficult relating the selected descriptors to biological activity [41]. These seemingly contradictory features between interpretability and predictivity can be solved by a novel two-QSAR approach [43] by incorporating the ML-based hierarchical support vector regression (HSVR) scheme [44] and the linear PLS scheme. Herein, this study was aimed at predicting the *K*_p_ values based on the ex vivo human skin permeability data for facilitating drug discovery by means of the two-QSAR scheme.

## 2. Materials and Methods

### 2.1. Data Collection

A sound predictive model can only be developed based on the data samples with good quality [41]. An exhaustive search was conducted to retrieve skin permeability parameters from the public domain to maximize data collection and to diversify the chemical structure. Skin permeability, nevertheless, is sensitive to the assay conditions (*vide supra*). Accordingly, *K*_p_ values, which were measured by the same assay conditions, were collected after cautious examination to secure data consistency. The arithmetic average *K*_p_ value was taken to ensure better assay consistency if there were two, and more than two, *K*_p_ values available within a narrow range for a given compound. Finally, 96 compounds were compiled in this investigation from various sources [45,46,47,48,49,50,51,52,53,54,55,56,57,58,59,60,61,62,63,64,65] and their corresponding logarithm *K*_p_, *viz*. p*K*_p_, values, simplified molecular input line entry system (SMILES) strings, Chemical Abstracts Service (CAS) registry numbers, and references to the literature are listed in Appendix A.

### 2.2. Molecular Descriptors

Full geometry optimization was carried out for all selected molecules to find the most stable conformation by means of the density functional theory (DFT) B3LYP method and the basis set of 6-31G (*d*,*p*) using the *Gaussian* package (Gaussian, Wallingford, CT). To mimic the experimental environment, the polarizable continuum model (PCM) [66,67] was employed to address the solvent system. The molecular electrostatic potential (MEP) [68] was adopted to calculate atomic charges, which are associated with dipole moments (*μ*). The energies of frontier orbitals, namely the highest occupied molecular orbital energy (HOMO) and the lowest unoccupied molecular orbital energy (LUMO), *μ*, and the largest absolute component of *μ* (|*μ*|_max_) were also obtained from the optimization calculations.

*Discovery Studio* (BIOVIA, San Diego, CA, USA) and *E-Dragon* (available at the website http://www.vcclab.org/lab/edragon/, accessed on 20 March 2020.) were employed to enumerate 1D-, 2D-, and 3D-descriptors that can be classified as topological, electronic, thermodynamic, structural, spatial, and E-state indices. *XLOGP3* of *SwissADME* (available at the website http://www.swissadme.ch/index.php, accessed on 20 March 2020.) was adopted to compute the logarithm of *n*-octanol–water partition coefficient at pH 7.4, namely log *P*. The scheme modified by Muehlbacher et al. [69] was used to calculate the cross-sectional area (CSA) due to its implication in membrane permeability [70,71]. The selected molecules were categorized into four different ion classes by their p*K*_a_ values, namely zwitterion, base, acid, and neutral ions. [72] More specifically, compounds with only one p*K*_a_ value are termed neutral ions; compounds with no p*K*_a_ values larger than 7 are coined acidic ions; compounds with no p*K*_a_ values smaller than 7 are coined basic ions; and compounds with the strongest acidic p*K*_a_ values larger than 7 and the strongest basic p*K*_a_ values smaller than 7 are called zwitterion ions.

### 2.3. Descriptor Selection

Initially, those descriptors missing more than one value were removed from the pool, followed by deleting those exhibited little or no discrimination among all data samples. The Spearman’s matrix, which calculated the correlations between descriptors, was constructed. It has been recommended by Topliss and Edwards that those descriptors showing the values of *r*^2^ > 0.80 should be dropped in order to decrease the chance of spurious correlations [73]. However, a more restricted value of *r*^2^ > 0.64 was adopted in this investigation.

It is not uncommon to observe that some descriptors span substantially broader ranges than the other due to their distinct nature. As such, it is of necessity to reduce the chances in which those descriptors with larger ranges override those with smaller ranges, by transferring descriptors into a more consistent range [74]. Accordingly, descriptor normalization was carried out by centering and scaling
(2)$${\hat{\chi }}_{ij}=\left(x_{ij}-\langle x_{j}\rangle \right)/{\left[\sum_{i=1}^{n}{\left(x_{ij}-\langle x_{j}\rangle \right)}^{2}/\left(n-1\right)\right]}^{1/2}$$
where $x_{ij}$ stands for the *j*-th original descriptor of the *i*-th compound with an average value of < *x_j_*> for *n* compounds, and ${\hat{\chi }}_{ij}$ symbolizes the normalized descriptor.

The selected descriptors play a vital role in the QSAR model performance [75]. A two-stage scheme was employed in this investigation, in which the genetic function approximation (GFA) implemented in the QSAR module of *Discovery Studio* was initially adopted mainly due to its effectiveness and efficiency [76]. Further descriptor selection was executed by the recursive feature elimination (RFE) scheme, in which the model was continually produced by all but one descriptor. The descriptors were then ranked according to their predictive performance and the one with the least contribution was purged first [77].

### 2.4. Dataset Selection

It is a common practice to recognize and remove the outliers from the sample pool for model development [78]. Nevertheless, outliers were identified and intended to challenge the robustness of developed models in this study [79]. As such, all selected molecules were projected into the chemical space constituted by principal components (PCs) using the function *Diverse Molecules* in the *Principal Component Analysis* (PCA) module of *Discovery Studio*. The outliers were then discovered by inspecting molecular distribution in the chemical space [80]. The *Diverse Molecules* function in the *Library Analysis* module of *Discovery Studio* was employed to randomly partition the rest of the molecules into the training set and test set to produce and verify the developed models, respectively, by a ca. 4:1 ratio as advocated [80]. In addition, the chemical and biological distributions in the training and test set were cautiously inspected since it has been suggested by Golbraikh et al. that chemical similarity and biological similarity in both data sets should be preserved in order to produce a sound theoretical model [81].

### 2.5. Hierarchical Support Vector Regression

Initially, Vapnik et al. proposed a support vector machine (SVM) for classification and further modified SVM for regression, termed as support vector regression (SVR) [82]. SVR functions by nonlinearly transferring the input into a higher-dimension space where linear regression is conducted [83]. Unlike traditional regression schemes, which develop predictive models by lowering the training error, SVR, conversely, takes into consideration both the training error and model complexity. Thus, it is not a surprise to observe that SVR performs much better than the traditional regression algorithms that predominantly can be related to its valuable features, namely independence of dimension, frugal number of freedom, excellent generalization property, universal optimum, and effortless realization [84].

The innovative hierarchical support vector regression (HSVR) scheme was initially derived by Leong et al. based on SVR [44]. One of the most unique and advantageous features of HSVR is its ability to resolve the seeming conflict between a global model and a local model, *viz*. the coverage of applicability domain (AD) vs. the level of predictivity [85]. More notably, the excellent performance of HSVR can be illustrated by a number of studies [7,43,44,86,87,88,89].

The principle and realization of HSVR have been reported in detail elsewhere, and the architecture of the HSVR scheme can be illustrated by Figure 1 of Leong et al. [44]. Principally, an SVR ensemble (SVRE) is compiled by assembling a pool of SVR models, which are generated based on different descriptor combinations to represent various local models with distinct ADs. The *svm*-*train* module in *LIBSVM* (software available at http://www.csie.ntu.edu.tw/~cjlin/libsvm/, accessed on 2 April 2020.) was employed to produce SVR models from those molecules in the training set with assorted descriptor selections and SVR run parameters. The *svm*-*predict* module in *LIBSVM* was chosen to verify the derived SVR models using the test samples. Radial basis function (RBF) was set to be the kernel due to its simplicity and considerable functionality [90]. Both *ν*-SVR and *ε*-SVR regression models were taken into account. The SVR runtime conditions including *ε* in *ε*-SVR and *ν* in *ν*-SVR, the kernel width *γ*, and cost *C* were automatically investigated by an in-house Perl script to do a systemic grid search.

The construction of SVRE was ruled by the principle of Occam’s razor, in which simpler models are preferred to the more complex ones [91]. This parsimony principle was also applied to the descriptor selection and construction of SVRE in that the number of selected descriptors and the number of ensemble members were kept as minimal as possible [92]. For instance, no three-member ensembles were to be constructed prior to the successful development of a two-member ensemble.

### 2.6. Partial Least Square

Partial least square is a general regression scheme that can develop models based on collinear descriptors. PLS can proceed the model development even in the case where there are more descriptors than the observables that makes it different from the other linear regression schemes [93]. The cross-validation scheme is often adopted to test the complexity of developed PLS models in order to reduce the chance correlations [94]. The PLS model was derived using the *Partial Least Square* module of *Discovery Studio*.

### 2.7. Predictive Evaluation

The prediction deviation or residual between the observed value $\left(y_{i}\right)$ and the predicted value (${\hat{y}}_{i}$) for the *i*-th molecule was evaluated by the following equation
(3)$$\Delta_{i}=y_{i}-{\hat{y}}_{i}$$

In addition, standard deviation (*s*), maximum residual (∆_Max_), root mean square error (RMSE), and mean absolute error (MAE) in a dataset with *n* molecules were calculated by the following equations
(4)$$\mathrm{RMSE}=\sqrt{\sum_{i=1}^{n}\Delta_{i}^{2}/n}$$
(5)$$\mathrm{MAE}=\frac{1}{n}\sum_{i=1}^{n}\left|\Delta_{i}\right|$$

The squared correlation coefficients, *viz*. *r*^2^ and *q*^2^ in the training set and external dataset, respectively, were computed to evaluate the developed models by the following equation.
(6)$$r^{2},\;q^{2}=1-\frac{\sum_{i=1}^{n}\Delta_{i}^{2}}{\sum_{i=1}^{n}{\left(y_{i}-\langle \hat{y}\rangle \right)}^{2}}$$
where $\langle \hat{y}\rangle$ symbolizes the average predicted value of *n* samples in the dataset.

The 10-fold cross-validation scheme was adopted to internally validate the derived model to produce the squared correlation coefficient $q_{\mathrm{CV}}^{2}$. The built models were subjected to the *Y*-scrambling test for further internal validation [95], which were executed by randomly reshuffling the log *K*_p_ values, followed by reapplied them to the previously built model without modifying the descriptors. The reshuffling process was carried out 25 times as suggested [95].

Furthermore, various derivative versions of *r*^2^ recommended by Ojha et al. [96] were also computed
(7)$$r_{m}^{2}=r^{2}\left(1-\sqrt{\left|r^{2}-r_{o}^{2}\right|}\right)$$
(8)$${r^{\prime }}_{m}^{2}=r^{2}\left(1-\sqrt{\left|r^{2}-{r^{\prime }}_{o}^{2}\right|}\right)$$
(9)$$\langle r_{m}^{2}\rangle =\left(r_{m}^{2}+{r^{\prime }}_{m}^{2}\right)/2$$
(10)$$\Delta r_{m}^{2}=\left|r_{m}^{2}-{r^{\prime }}_{m}^{2}\right|$$
where the squared correlation coefficient $r_{o}^{2}$ and the slope of the regression line *k* were resulted from the regression line (predicted vs. observed values) without offset at the intersection, whereas ${r^{\prime }}_{o}^{2}$ was obtained from the regression line (observed vs. predicted values) without offset at the intersection.

The predictivity of derived models were challenged by the external datasets to give rise to various squared correlation coefficients, namely $q_{F1}^{2}$, $q_{F2}^{2}$, and $q_{F3}^{2}$, as well as the concordance correlation coefficient (*CCC*) [87] using *QSARINS* [97,98]
(11)$$q_{F1}^{2}=1-\frac{\sum_{i=1}^{n_{\mathrm{EXT}}}\Delta_{i}^{2}}{\sum_{i=1}^{n_{\mathrm{EXT}}}{\left(y_{i}-\langle y_{\mathrm{TR}}\rangle \right)}^{2}}$$
(12)$$q_{F2}^{2}=1-\frac{\sum_{i=1}^{n_{\mathrm{EXT}}}\Delta_{i}^{2}}{\sum_{i=1}^{n_{\mathrm{EXT}}}{\left(y_{i}-\langle y_{\mathrm{EXT}}\rangle \right)}^{2}}$$
(13)$$q_{F3}^{2}=1-\frac{\sum_{i=1}^{n_{\mathrm{EXT}}}\Delta_{i}^{2}/n_{\mathrm{EXT}}}{\sum_{i=1}^{n_{\mathrm{EXT}}}{\left(y_{i}-\langle y_{\mathrm{TR}}\rangle \right)}^{2}/n_{\mathrm{TR}}}$$
(14)$$CCC=\frac{2\sum_{i=1}^{n_{\mathrm{EXT}}}\left(y_{i}-\langle y_{\mathrm{EXT}}\rangle \right)\left({\hat{y}}_{i}-\langle {\hat{y}}_{\mathrm{EXT}}\rangle \right)}{\sum_{i=1}^{n_{\mathrm{EXT}}}{\left(y_{i}-\langle y_{\mathrm{EXT}}\rangle \right)}^{2}+\sum_{i=1}^{n_{\mathrm{EXT}}}{\left({\hat{y}}_{i}-\langle {\hat{y}}_{\mathrm{EXT}}\rangle \right)}^{2}+n_{\mathrm{EXT}}{\left(\langle y_{\mathrm{EXT}}\rangle -\langle {\hat{y}}_{\mathrm{EXT}}\rangle \right)}^{2}}$$
where $\langle y_{\mathrm{TR}}\rangle$ stands for the averaged observed values of $n_{\mathrm{TR}}$ samples in the training set, $\langle y_{\mathrm{EXT}}\rangle$ and $\langle {\hat{y}}_{\mathrm{EXT}}\rangle$ are the averaged observed and predicted values of $n_{\mathrm{EXT}}$ samples in the external set, respectively.

Most significantly, only models that can meet the most stringent criteria altogether asserted by Golbraikh et al. [81], Ojha et al. [96], Roy et al. [99], and Chirico and Gramatica [100].

Moreover, various modified squared correlation coefficients *r*^2^ were also computed
(15)$$\;r^{2},\;q_{\mathrm{cv}}^{2},\;q^{2},\;q_{F1}^{2},\;q_{F2}^{2},\;q_{F3}^{2}\geq 0.70$$
(16)$$\left|r^{2}-q_{\mathrm{cv}}^{2}\right|<0.10$$
(17)$$\left(r^{2}-r_{o}^{2}\right)/r^{2}<0.10\;\mathrm{and}\;0.85\leq k\leq 1.15$$
(18)$$\left|r_{0}^{2}-{r^{\prime }}_{0}^{2}\right|<0.30$$
(19)$$r_{m}^{2}\geq 0.65$$
(20)$$\langle r_{m}^{2}\rangle \geq 0.65\;\mathrm{and}\;\Delta r_{m}^{2}<0.20$$
(21)CCC≥0.85
where *r*^2^ in Equations (17)−(20) represents *r*^2^ in the training set and *q*^2^ in the external set.

## 3. Results

### 3.1. Data Partition

Of all selected molecules, 73 and 18 molecules were arbitrarily designated as the training samples and the test samples, respectively, yielding an approximate 4:1 ratio as suggested [80]. Figure 1 exhibits the projection of all selected molecules. As displayed, both data sets showed great degrees of similarity in chemical space, which was constructed by the span of the first three PCs that characterized 99.44% of the variance in the original data. In addition, the great levels of the biological and chemical similarity between the training samples and test samples can also be exemplified by those graphs in Appendix A, which features the histograms of log *K*_p_, molecular weight (MW), molecular volume (*V*_m_), *n*-octanol−water partition coefficient (log *P*), number of hydrogen-bond acceptors (HBA), and number of hydrogen-bond acceptors (HBD) in the form of density for those samples in the training set and test set. Accordingly, the nondiscriminatory data partition can be further assured [101].

Conversely, those designated outliers are distinctly positioned from tall the other samples, namely the training and test samples, as exhibited in Figure 1. The biological and chemical dissimilarity between the outliers and the others can be further manifested by the histograms shown in Appendix A. In fact, the dissemblance between outliers and the others can be realized by the fact that the outliers consisted of a greater number of rotatable bond (*N*_rot_ ≥ 8) or a greater number of oxygen (*N*_O_ ≥ 11).

### 3.2. HSVR

Of all derived SVR models based on different descriptor combinations as well as runtime parameters, three SVR models, termed as SVR A, SVR B, and SVR C, were assembled to constitute the SVR ensemble, whose predictions, in turn, were treated as the input of another SVR to build the HSVR model. Appendix A records the optimal runtime parameters of SVR A, SVR B, SVR C, and HSVR.

It is of interest to note SVR A, SVR B, and SVR C simultaneously adopted two descriptors with different selections as listed in Table 1. HSVR generally produced the medium prediction errors as compared with SVR A, SVR B, and SVR C (Appendix A). Additionally, it can be detected from Figure 2 and Figure 3, which exhibit the scatter plots of observed vs. predicted log *K*_p_ values in the training set and test set, respectively, that the lengths between the predictions by HSVR and regression line were between those yielded by those SVR models in the ensemble.

However, it is not uncommon to find that HSVR produced the smallest residuals for some predictions. For instance, the prediction of **51** (benzoic acid) by HSVR gave rise to an absolute residual of 0.05, whereas SVR A, SVR B, and SVR C yielded the absolute residuals of 0.11, 0.13, and 0.14, respectively. It can be concluded from those statistical parameters listed in Table 2 and Table 3 that HSVR functioned better than those SVR models in the ensemble in the training set and test set. Furthermore, HSVR delivered the highest *r*^2^ (0.93) and $q_{\mathrm{CV}}^{2}$ (0.90) and the lowest Δ_Max_ (1.11), MAE (0.21), and RMSE (0.32) in the training set, suggesting the superior performance of HSVR in the training set. It is of interest to note that HSVR delivered a marginal difference between *r*^2^ and $q_{\mathrm{CV}}^{2}$ (0.03), whereas the smallest difference between both parameters was produced by SVR B with a value of 0.14 for all SVR models in the ensemble, denoting that HSVR was well trained and those SVR models in the ensemble were less well trained. However, little chance correlation was associated with all SVR models as assured by their nearly zero values of $\langle r_{s}^{2}\rangle$ (0.01) upon the *Y*-scrambling test [95].

SVR B and HSVR showed excellent performance in the test set as manifested by those statistical metrics listed in Table 3 as well as little performance difference between the training set and test set. For instance, SVR B and HSVR produced various *q*^2^ parameters of 0.88–0.93 and 0.84–0.92, respectively. In fact, SVR B marginally executed better than HSVR in every aspect except Δ_Max_ (0.83 vs. 0.66). Conversely, SVR C displayed substantial performance deteriorations from the training set to the test set as exemplified by the significant differences between *r*^2^ (0.82) in the training set and *q*^2^ (0.38−0.50) in the test set.

Figure 4 shows the scatter plot of the observed log *K*_p_ vs. the log *K*_p_ values predicted by the derived models in the outlier and the corresponding statistic metrics are listed in Table 4. It can be a misconception to assume that SVR B functioned better than HSVR in the outlier set by the parameter *q*^2^ (0.93 vs. 0.86). Nevertheless, the other statistic metrics, namely $q_{F1}^{2}$ − $q_{F3}^{2}$ and *CCC*, indicate otherwise. In fact, those SVR models in the ensemble unanimously generated extraordinary absolute deviations when applied to sucrose (**96**) with the values of 4.78, 2.21, and 2.88, whereas HSVR only delivered a negligible value of 0.04 for the same compound (Appendix A). Consequently, none of the SVR models in the ensemble generated positive $q_{F3}^{2}$. HSVR, conversely, gave rise to positive statistics metrics, the least difference between *r*^2^ and various *q*^2^ values, and the smallest Δ_Max_. MAE, *s*, and RMSE, suggesting that HSVR was very insensitive to the outliers that, in turn, made HSVR the most practically favorable due to its robust nature [79]. Accordingly, it is of necessity to construct an HSVR model that can be predictive as well as robust for various chemotypes of molecules.

### 3.3. PLS

The linear PLS model was derived by simultaneously compiling those descriptors selected by the SVR models in the SVRE (Table 1). The prediction results of the samples in the training set, test set, and outlier set are listed in Appendix A, and the associated statistical evaluations in three data sets are summarized in Table 2, Table 3 and Table 4, respectively.
log *K*_p_ = −2.62974 + 1.26972 × log *P* − 0.55661 × *V*_m_ − 0.554268 × ^0^*χ*
−0.076344 × Jurs_PPSA_1(22)

The PLS model yielded an *r*^2^ value of 0.80 in the training set, which is slightly smaller than those produced by SVR A, SVR B, SVR C, and HSVR. Nevertheless, it can be observed from Figure 3 that most of the points predicted by PLS generally had the largest distances from the regression line as compared with SVR A, SVR B, SVR C, and HSVR.

However, PLS produced a *q*^2^ value of 0.74 in the outlier set, which is better than those obtained by SVR A and SVR C (0.64 and 0.10). The *CCC* value calculated by PLS is even better than all those SVR models in the ensemble. The prediction of sucrose by PLS also presented an exceptionally large absolute error with a value of 4.33, which is similar to the observation found in SVR A, SVR B, and SVR C (*vide supra*). In addition, PLS also produced negative $q_{F1}^{2}$ − $q_{F3}^{2}$ values in the outlier set. As such, it can be asserted by all parameters listed in Table 4 that HSVR outperformed PLS in the outlier set as well.

### 3.4. Predictive Assessment

It can be discovered from Figure 5, which exhibits the scatter plot of the residuals vs. the log *K*_p_ values predicted by both theoretical models for the molecules in those three data sets, namely training set, test set, and outlier set, that the residuals generated by HSVR were generally equally situated on both sides of *x*-axis along with the prediction range in those three datasets, depicting the fact that little systematic errors were related with HSVR. As such, HSVR yielded the mean errors of −0.01, 0.05, and −0.26 in the training set, test set, and outlier set, respectively. PLS, similarly, produced small average errors in the training and test (0.00 and 0.03). However, the prediction of sucrose by PLS led to an unusual average error of 1.02 in the outlier set that can be further manifested by the PLS regression line shown in Figure 4 since the distinction between the PLS regression line and the ideal regression was pronounced, suggesting that a systematic error was associated with PLS in the outlier set.

The validation results are listed in Table 5 when HSVR and PLS were subjected to those criteria displayed in Equations (13)–(19). It is obvious to discover that HSVR not only produced the largest statistic parameters but also completely satisfied the most stringent requirements, whereas PLS showed substantial variations among those statistic parameters and failed to fulfill all of the criteria. Accordingly, it can be unequivocally concluded that the performance of HSVR is superior to that of PLS.

### 3.5. Mock Test

To imitate their applications in the real world, the derived HSVR and PLS models were applied to those compounds assayed by Soriano-Meseguer et al. [102], of which, 23 were also included in this investigation, providing a good way to calibrate the challenge system. Nevertheless, Soriano-Meseguer et al. measured the effective permeability coefficient (*P*_e_) based on the skin-PAMPA system. The discrepancies in the assay system and measurement can create data heterogenicity once the assay results from the skin-PAMPA system are pooled into the data collection [41]. Subsequently, the relationship between both different assay systems (log *P*_e_ vs. log *K*_p_) was initially constructed and examined based on those 23 common compounds and the resulted scatter plot is shown in Figure 6. It can be discovered that both assay systems were reasonably correlated with each other with an *r* value of 0.78, suggesting that it is plausible to adopt the data measured by Soriano-Meseguer et al. to challenge the derived HSVR and PLS models.

The predicted results of those 23 novel compounds in the mock tests are listed in Appendix A and displayed in Figure 7. It can be observed that HSVR gave rise to an *r* value of 0.71 between observed log *P*_e_ and predicted log *K*_p_, suggesting that HSVR can almost replicate the experimental measurements. PLS, conversely, yielded an extremely small value of 0.38, denoting the fact that PLS cannot reproduce the assay data. Accordingly, it can be asserted that HSVR outperformed PLS in the mock test.

### 3.6. Comparison with Skin Permeation Calculator

The US Center for Disease Control and Prevention (CDC) has developed *Skin Permeation Calculator* (SPC) (available at the website: https://www.cdc.gov/niosh/topics/skin/skinpermcalc.html, accessed on 9 September 2021) to predict skin permeation. It is of interest to compare SPC with PLS and HSVR by applying SPC to all of compounds enrolled in this investigation. The predicted results by SPC are listed in Appendix A and the statistic evaluations are listed in Table 6. Of all 96 compounds selected in this study, only 77 compounds, or 80%, could be predicted by SPC, suggesting that the applicability of SPC is limited when compared with PLS and HSVR.

Furthermore, SPC showed the worst performance as manifested by those statistical parameters listed in Table 6. PLS executed better than SPC, which can be plausibly attributed to the better descriptor selection as well as better descriptor enumeration. HSVR unequivocally performed better than PLS, indicating the superior architecture of HSVR scheme. In addition, those metrics calculated by HSVR in Table 6 did not vary a lot from those in different data sets (Table 2, Table 3 and Table 4), suggesting the performance stability of HSVR in different compounds.

## 4. Discussion

Skin permeation can take place through a series of processes, in which molecules must penetrate through various skin layers before they can reach the body circulation system (*vide supra*). As such, skin permeability is governed by various factors, of which log *P* and molecular weight (MW) are the two most frequently adopted descriptors that can be manifested by the predictive model developed by Potts and Guy [24], in which only both descriptors were used.

In fact, the significance of log *P* in skin permeability can be realized by the fact that all of the SVR models in the ensemble unanimously adopted this descriptor and can be further supported by the largest absolute weight (1.27) given by PLS (Equation (22)) among all of the selected descriptors. Furthermore, it has been observed by Potts and Guy that log *K*_p_ linearly increases with the increase of log *P*. Nevertheless, the correlation between log *P* and log *K*_p_ was only 0.42 for all of the molecules selected in this study (Table 1), which, actually, is similar to the value observed by Chen et al. (*r* = 0.467) [32]. This inconsistency is plausibly attributed to the fact that Potts and Guy selected the compounds of log *P* < 4 only, whereas this study and Chen et al. included some compounds of log *P* ≥ 4. More importantly, it can be observed from Figure 8, which displays the average log *K*_p_ for each histogram bin of log *P* for all selected molecules, that log *K*_p_ increased with log *P* initially and then decreased once log *P* ≥ 4, leading to an apparently bi-linearity between log *K*_p_ and log *P*.

This intricate reliance can be realized by the fact that it is easier for the more hydrophobic permeants to approach the skin lipid bilayer, which is hydrophobic *per se* [20]. Conversely, it will be harder for those too hydrophobic permeants to escape skin lipid bilayer or even retain in the skin layers without significant penetration. As such, the collective *r* is reduced once both the positive and negative *r* values are taken into account. It is plausible to expect that such bi-linearity cannot be properly addressed by linear models, whereas this nonlinearity can be appropriately handled by ML schemes provided that the other descriptors are properly selected.

Potts and Guy have adopted the descriptor MW to render the size impact on the skin permeability [24]. In fact, most published in silico models also have selected MW as the size-related descriptor. Nevertheless, none of the SVR models in the ensemble included MW and yet SVR A enrolled the descriptor molecular volume (*V*_m_). This divergence can be justified by the fact that MW was highly correlated with *V*_m_ with an *r* value of 0.96 for all of the molecules compiled in this study, suggesting that it is plausible to replace MW by *V*_m_ as a size-related descriptor. This justification, actually, is also consistent with the postulation made by Wilschut et al. that the molecular size can be better denoted by *V*_m_ when taking into account the electron-density distributions [103]. For instance, the steric isomers have the same MW, whereas their *V*_m_ values are different, indicating that MW cannot show the distinction between both steric isomers and *V*_m_ is a better way to render the size factor. As such, the empirical observation unequivocally indicated that models with the selection of *V*_m_ performed better than those with the selection of MW (data not shown) that, additionally, can be partially attributed to the fact that *V*_m_ was enumerated based the geometry that was fully optimized by the more sophisticated DFT with the selection of a descent basis set along with the consideration of a solvent effect.

The PLS placed a negative weight to *V*_m_ (Equation (22)) that is similar to the other published models, which unanimously gave negative coefficients to MW. The reverse relation between molecular size and skin permeability can be plausibly explained by the fact that molecular size is the most critical factor in demining the solute flux amounts through the epidermis since smaller solute molecules tend to have higher possibilities to enter the SC pores and, consequently, across the SC pores and lipid lamellar layers faster [104].

It is unusual to observe that SVR B adopted the descriptor ^0^χ, which depicts the molecular connectivity index of order zero, since none of the published in silico models has selected this descriptor. Nevertheless, it can be observed from Figure 9, which exhibits *V*_m_ versus ^0^χ, that *V*_m_ and ^0^χ were extremely correlated with each other (*r* = 0.98), suggesting that ^0^χ can be another descriptor to describe the size factor in skin permeability. The over-training issue was not applicable in this study since SVR A and SVR B recruited *V*_m_ and ^0^χ, respectively, *viz*. no simultaneous selection of both descriptors by any SVR model in the ensemble. The significance of ^0^χ in skin permeability can be manifested by the weight given by PLS, which is very similar to the one associated with *V*_m_ (−0.554268 vs. −0.55661). More importantly, the empirical operations have disclosed that HSVR based on this descriptor combination executed better than the others (data not shown) plausibly as a result of the descriptor–descriptor interaction [7]. Any other linear or nonlinear ML-based QSAR methods, contrarily, cannot properly address such paradoxical descriptor selections.

It is of interest to note the selection of partial positive surface area (Jurs_PPSA_1) by SVR C since it has never been included by any published in silico models. Nevertheless, it has been observed that polar surface area (PSA) plays a significant role in distinguishing between the substrates and non-substrates of P-glycoprotein (P-gp) [87]. It has been found that P-gp can be expressed in the human skin [105]. In contrast to the intestine and blood–brain barrier (BBB), the efflux transporter P-gp in the skin plays an influx role by transporting substrates from the surface into the dermis [106]. As such, the descriptor Jurs_PPSA_1, which is a modified version of PSA, was adopted in this study with better model performance (data not shown). Compounds selected in this study were further classified as P-gp substrates and non-P-gp substrates using *admetSAR* (available at http://lmmd.ecust.edu.cn/admetsar2/, accessed on 17 September 2021.) to investigate the Jurs_PPSA_1 impact on the skin permeability. The results are shown in Figure 10, which displays the plot of log *K*_p_ versus Jurs_PPSA_1 for those P-gp substrates and non-P-gp substrates along with their associated regression lines. It can be observed that Jurs_PPSA_1 was substantially associated with log *K*_p_ with an *r* value of 0.96 for P-gp substrates, whereas there was a negative correlation between log *K*_p_ and Jurs_PPSA_1 for non-P-gp substrates, suggesting that PSA can facilitate the influx of P-gp substrate that, in turn, can enhance the skin permeation consequently.

PSA can also represent molecular polarity [87]. Abraham et al. has adopted a molecular polarity-related descriptor to describe its impact on skin permeability [29]. The negative coefficient of Jurs_PPSA_1 given by PLS (−0.076344) as well as the negative weight associated with the polar descriptor in the model developed by Abraham et al. unequivocally indicate the reverse relationship between PSA/polarity and skin permeability. Additionally, larger PSA or dipole will result in stronger interactions between solute and solute as well as between solute and solvent, increasing higher desolvation energy when they approach the skin lipid bilayer [7].

The rather small *r* value (−0.34) between log *K*_p_ and Jurs_PPSA_1 for non-P-gp substrates can be presumably attributed to the different permeation routes for molecules with different polarities (*vide supra*) as well the nature of solute−solute and solute−solvent interactions. As such, Jurs_PPSA_1 plays a profound role in skin permeation since it can simultaneously enhance and reduce skin permeation depending on the nature of the permeant and such a contradictory feature cannot be properly depicted by any traditional linear model. HSVR, conversely, can correctly render such complicated relationship.

It has been observed the neutral compounds are more permeable in the human colon carcinoma cell layer (Caco-2) and parallel artificial membrane permeability assay (PAMPA) system [7,89]. It is of interest to investigate that if neutral compounds have higher permeability values in the ex vivo skin permeability model as compared with the other ion classes. All of the molecules enlisted in this study were categorized into four ion classes according to their p*K*_a_ values. It can be found from Figure 11, which demonstrates the box plot of the log *K*_p_ minimum, maximum, mean, median, the 25th percentile, and the 75th percentile for each ion class, that the log *K*_p_ values of neutral compounds are larger than the other ion classes, whereas that of basic compounds are statistically lower than the others, suggesting that neutral compounds are more permeable, which is consistent with the observation made by the Caco-2 and PAMPA systems, and basic compounds are less likely to penetrate through skin.

## 5. Conclusions

Topical and transdermal drug delivery is an effective, safe, and preferred route of drug administration. Skin permeability plays a pivotal role in drug discovery and development. This investigation used a novel two-QSAR scheme by collectively incorporating hierarchical support vector regression as well as partial least square to predict log *K*_p_ values based on the ex vivo skin permeability values compiled from the literature. The built HSVR model exhibited exceptional performance in three data sets, namely the training set, test set, and outlier set, whereas PLS modestly functioned in those three data sets. Various statistical evaluations and validation assessments asserted the accuracy and predictivity of HSVR. The mock test further asserted the practical application of HSVR, whereas PLS failed to deliver the satisfactory performance for the mock test. It is plausible to assure that the unique architectures of HSVR that can concurrently retain the advantageous features of a local model and a global model, namely broader applicability domain as well as greater predictivity, respectively, make substantial contribution to its superior performance, generalization ability, and robustness. PLS, which is a linear model, managed to reveal the interpretable relevance between selected descriptors and permeability that otherwise cannot be done any “black box” approaches. Both models also displayed good performance in qualitative prediction. Accordingly, the synergy between predictive HSVR and interpretable PLS can be useful to predict the skin permeability and to render the mechanisms associated with skin permeation, respectively. More importantly, this investigation has paved the way to predict the in vivo human skin permeability of hit and lead compounds in the future.

## Figures and Tables

**Figure 1 pharmaceutics-14-00961-f001:**
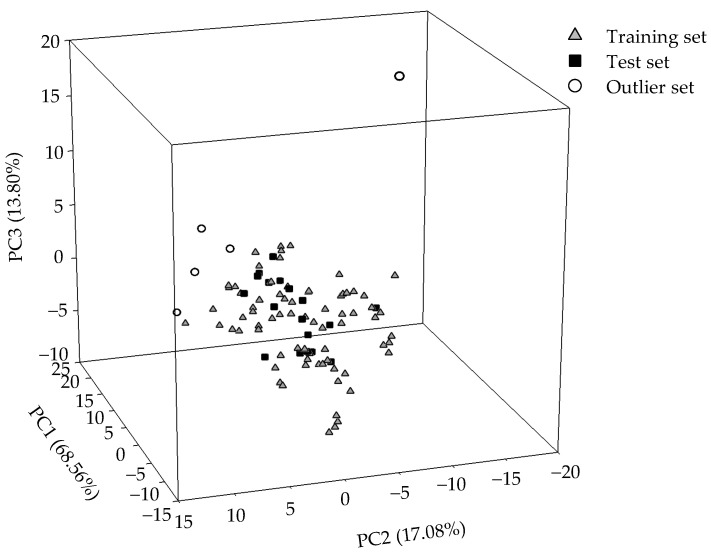
Molecular distribution for the selected molecules in the training set (gray triangle), test set (solid square), and outlier set (open circle) in the chemical space spanned by three PCs.

**Figure 2 pharmaceutics-14-00961-f002:**
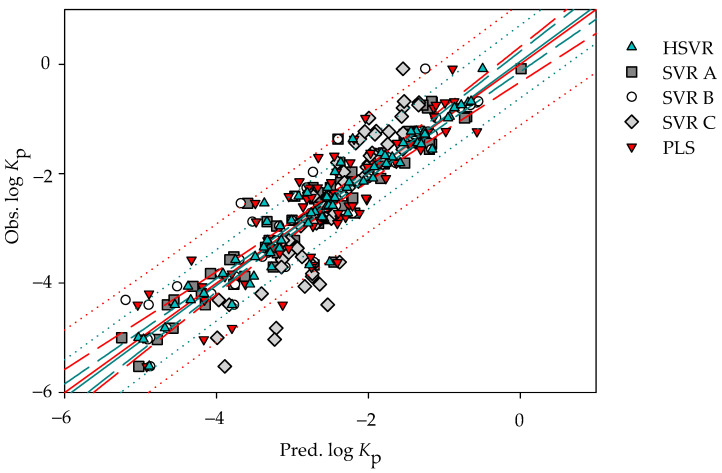
Observed log *K*_p_ vs. the log *K*_p_ predicted by SVR A (solid square), SVR B (open circle), SVR C (gray diamond), HSVR (green triangle), and PLS (red inverted triangle) for the molecules in the training set. The green and red solid lines, dashed lines, and dotted lines correspond to the HSVR and PLS regressions of the data, 95% confidence intervals for the HSVR and PLS regressions, and 95% confidence intervals for the prediction, respectively.

**Figure 3 pharmaceutics-14-00961-f003:**
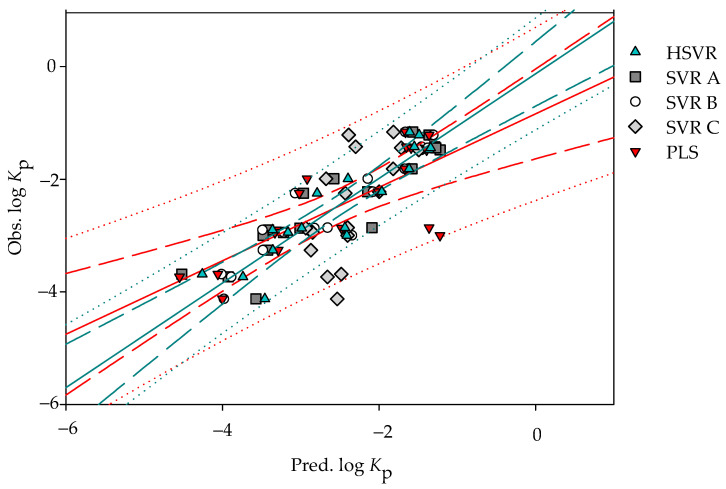
Observed log *K*_p_ vs. the log *K*_p_ predicted by SVR A (solid square), SVR B (open circle), SVR C (gray diamond), HSVR (green triangle), and PLS (red inverted triangle) for the molecules in the test set. The green and red solid lines, dashed lines, and dotted lines correspond to the HSVR and PLS regressions of the data, 95% confidence intervals for the HSVR and PLS regressions, and 95% confidence intervals for the prediction, respectively.

**Figure 4 pharmaceutics-14-00961-f004:**
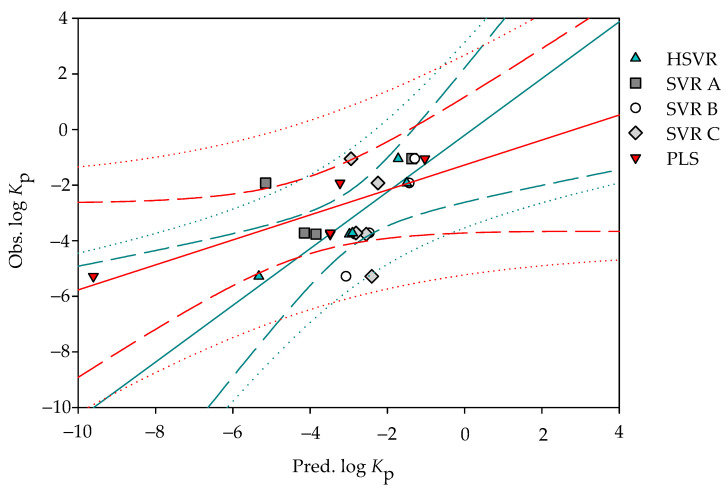
Observed log *K*_p_ vs. the log *K*_p_ predicted by SVR A (solid square), SVR B (open circle), SVR C (gray diamond), HSVR (green triangle), and PLS (red inverted triangle) for the molecules in the outlier set. The green and red solid lines, dashed lines, and dotted lines correspond to the HSVR and PLS regressions of the data, 95% confidence intervals for the HSVR and PLS regressions, and 95% confidence intervals for the prediction, respectively.

**Figure 5 pharmaceutics-14-00961-f005:**
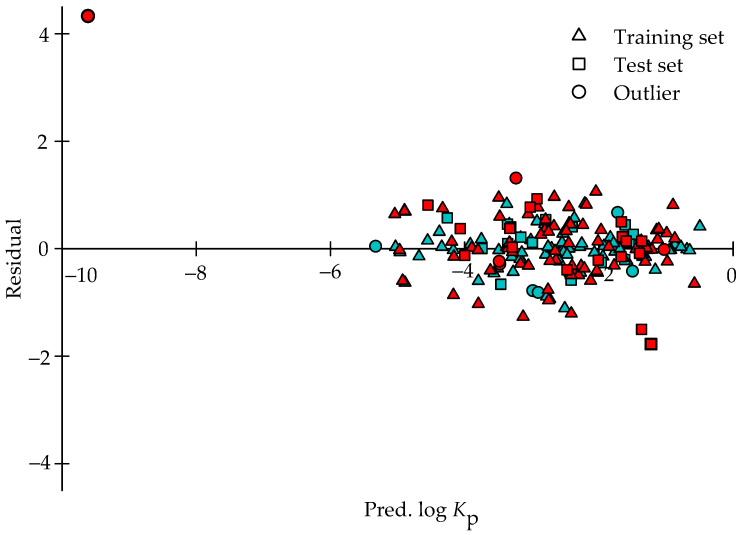
Residual vs. the log *K*_p_ values predicted by HSVR (green) and PLS (red) in the training set (square), test set (circle), and outlier set (triangle).

**Figure 6 pharmaceutics-14-00961-f006:**
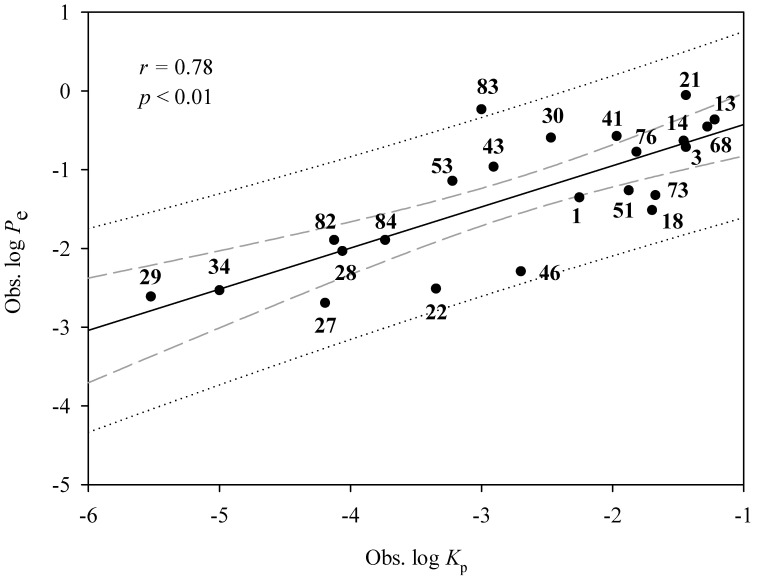
Observed log *P*_e_ vs. observed log *K*_p_ for the common compounds in the mock test. The solid line, dashed line, and dotted lines correspond to the mock test regression of the observed data, 95% confidence interval for the mock test regression, and 95% confidence interval for the observation, respectively.

**Figure 7 pharmaceutics-14-00961-f007:**
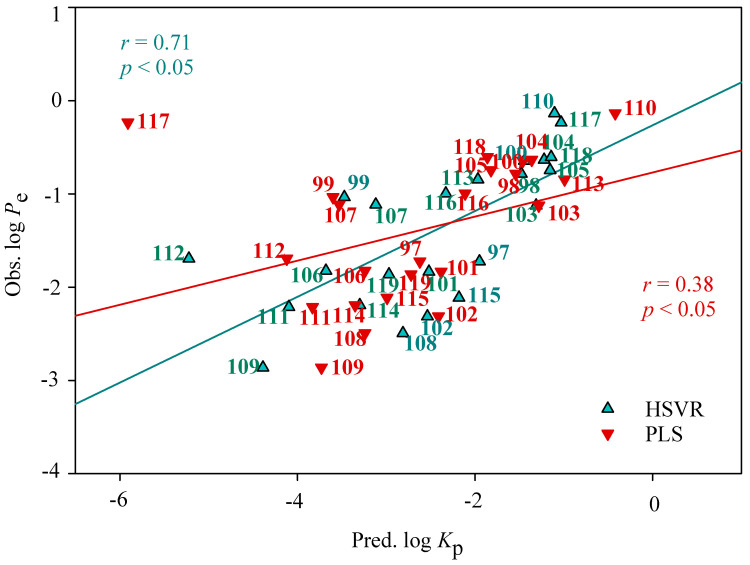
The observed log *P*_e_ values vs. the log *K*_p_ values predicted by HSVR (green triangle) and PLS (red inverted triangle), and their regression lines.

**Figure 8 pharmaceutics-14-00961-f008:**
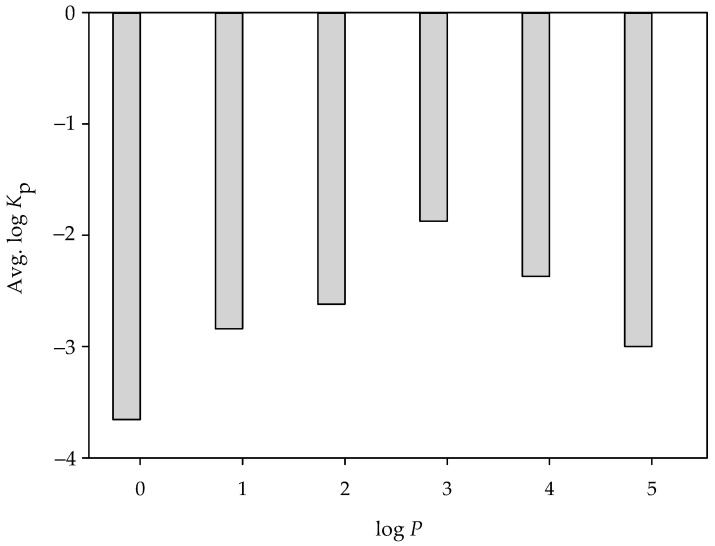
Histogram of average log *K*_p_ versus the distribution of log *P*.

**Figure 9 pharmaceutics-14-00961-f009:**
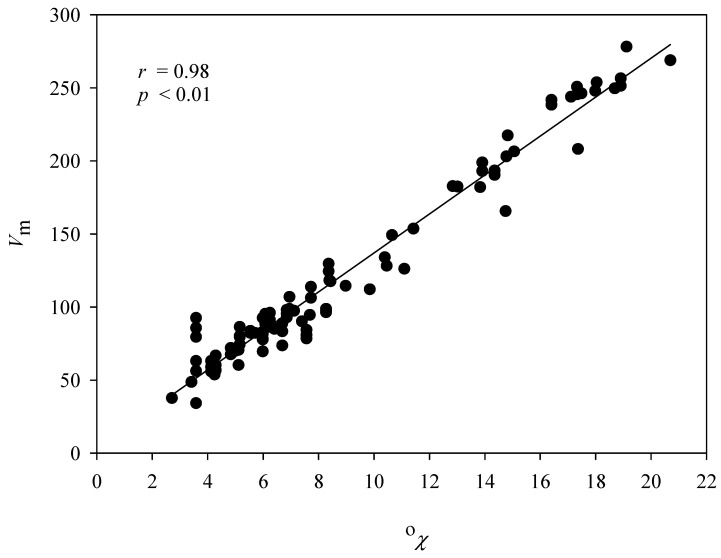
A plot of *V*_m_ versus ^0^χ.

**Figure 10 pharmaceutics-14-00961-f010:**
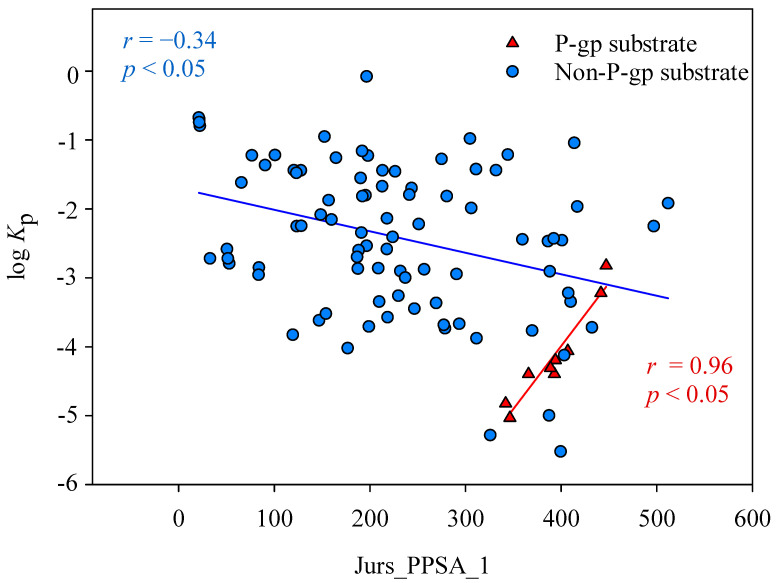
A plot of log *K*_p_ versus Jurs_PPSA_1 for those P-gp substrates (red triangle) and non-P-gp substrates (blue circle). The solid red and blue lines represent the P-gp regression data and non-substrate regression data, respectively.

**Figure 11 pharmaceutics-14-00961-f011:**
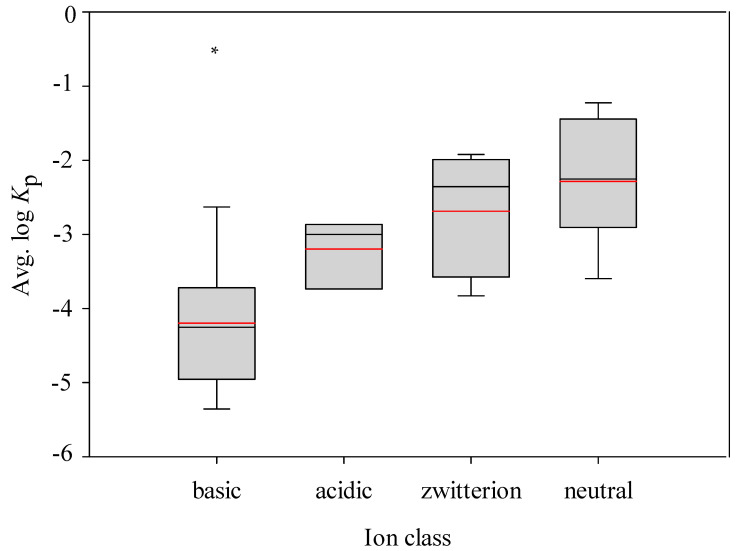
Box plot of log *K*_p_ values for four ion classes, where the boxes characterize the distribution of log *K*_p_ from the 25th to the 75th percentile, the black and red lines delineate the median and mean values, the whiskers denote the minimum and maximum values, and the asterisk specifies the significant difference between neutral and the other ion classes (*p* < 0.05).

**Table 1 pharmaceutics-14-00961-t001:** Descriptor selected as the input of SVR models in the ensemble, the correlation coefficient (*r*) with log *K*_p_, and their descriptions.

Descriptor	SVR A	SVR B	SVR C	*r*	Description
log *P*	x ^†^	x	x	0.42	Logarithm of the *n*-octanol−water partition coefficient
*V* _m_	x			−0.43	Molecular volume
^0^χ		x		−0.48	Molecular connectivity index of order zero
Jurs_PPSA_1			x	−0.43	Partial positive surface area

^†^ Selected.

**Table 2 pharmaceutics-14-00961-t002:** Statistic evaluations, namely squared correlation coefficient (*r*^2^), 10-fold cross-validation squared correlation coefficient ($q_{\mathrm{CV}}^{2}$), maximum error (Δ_Max_), mean absolute error (MAE), standard deviation (*s*), root mean square (RMSE), and $\langle r_{s}^{2}\rangle$ evaluated by SVR A, SVR B, SVR C, HSVR, and PLS in the training set.

	SVR A	SVR B	SVR C	HSVR	PLS
*r* ^2^	0.90	0.89	0.82	0.93	0.80
*q^2^_cv_*	0.57	0.75	0.60	0.90	0.90
Δ_Max_	1.17	1.17	1.86	1.11	1.27
MAE	0.29	0.27	0.48	0.21	0.41
*s*	0.23	0.29	0.49	0.24	0.33
RMSE	0.37	0.39	0.68	0.32	0.53
$\langle r_{s}^{2}\rangle$	0.01	0.01	0.01	0.01	0.01

**Table 3 pharmaceutics-14-00961-t003:** Statistic evaluations, namely *q*^2^, $q_{F1}^{2}$, $q_{F2}^{2}$, $q_{F3}^{2}$, *CCC*, Δ_Max_, MAE, *s*, and RMSE evaluated by SVR A, SVR B, SVR C, HSVR, and PLS in the test set.

	SVR A	SVR B	SVR C	HSVR	PLS
*q* ^2^	0.83	0.88	0.42	0.84	0.58
	0.76	0.86	0.39	0.83	0.42
$q_{F2}^{2}$	0.76	0.86	0.38	0.83	0.41
$q_{F3}^{2}$	0.86	0.92	0.64	0.90	0.66
*CCC*	0.90	0.93	0.50	0.92	0.75
Δ_Max_	0.84	0.83	1.59	0.66	1.78
MAE	0.37	0.25	0.53	0.31	0.50
*s*	0.24	0.23	0.48	0.20	0.49
RMSE	0.44	0.34	0.71	0.37	0.69

**Table 4 pharmaceutics-14-00961-t004:** Statistic evaluations, namely *q*^2^, $q_{F1}^{2}$, $q_{F2}^{2}$, $q_{F3}^{2}$, *CCC*, Δ_Max_, MAE, *s*, and RMSE evaluated by SVR A, SVR B, SVR C, HSVR, and PLS in the outlier set.

	SVR A	SVR B	SVR C	HSVR	PLS
*q* ^2^	0.64	0.93	0.10	0.86	0.74
$q_{F1}^{2}$	−1.60	0.41	−0.11	0.85	−0.59
$q_{F2}^{2}$	−2.00	0.32	−0.28	0.83	−0.84
$q_{F3}^{2}$	−3.79	−0.08	−1.04	0.73	−1.93
*CCC*	0.50	0.58	−0.09	0.91	0.64
Δ_Max_	4.78	2.21	2.88	0.82	4.33
MAE	1.77	1.02	1.45	0.55	1.24
*s*	2.12	0.77	0.98	0.32	1.80
RMSE	2.59	1.23	1.69	0.62	2.03

**Table 5 pharmaceutics-14-00961-t005:** Validation verification of HSVR and PLS based on prediction performance evaluated in the training set, test set, and outlier set.

	Training Set		Test Set		Outlier Set	
	HSVR	PLS	HSVR	PLS	HSVR	PLS
	0.93	0.80	0.84	0.45	0.86	0.51
*k*	1.01	1.00	0.97	0.94	1.08	0.66
${r^{\prime }}_{0}^{2}$	0.92	0.76	0.83	0.57	0.85	0.72
$r_{m}^{2}$	0.91	0.80	0.80	0.37	0.81	0.38
${r^{\prime }}_{m}^{2}$	0.85	0.64	0.75	0.52	0.80	0.63
$\langle r_{m}^{2}\rangle$	0.88	0.72	0.77	0.45	0.80	0.51
$\Delta r_{m}^{2}$	0.06	0.16	0.05	0.15	0.01	0.24
Equation (13)	x ^†^	x	x		x	
Equation (14)	x		N/A ^‡^	N/A	N/A	N/A
Equation (15)	x	x	x		x	
Equation (16)	x	x	x	x	x	x
Equation (17)	x	x	x		x	
Equation (18)	x	x	x		x	
Equation (19)	x	x	x		x	

^†^ Fulfilled; ^‡^ Not applicable.

**Table 6 pharmaceutics-14-00961-t006:** Statistical evaluations, namely, squared correlation coefficient (*r*^2^), maximal absolute residual (Δ_Max_), mean of absolute error (MAE), standard deviation (s), and root mean square error (RMSE), evaluated by HSVR, PLS, and SPC based on 78 compounds.

	HSVR	PLS	SPC
*r* ^2^	0.91	0.82	0.66
Δ_Max_	1.11	1.31	1.84
MAE	0.26	0.39	0.54
*s*	0.25	0.33	0.48
RMSE	0.36	0.51	0.73

## Data Availability

Not applicable.

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
