# Peer review of "In Silico Prediction of Skin Permeability Using a Two-QSAR Approach"

_pharmaceutics, 2022, doi:10.3390/pharmaceutics14050961_

Round 1
Reviewer 1 Report
Overall the month is well written and the results presented are satisfactory.
Author Response
Nothing needed to be done.
Reviewer 2 Report
The manuscript is interesting and the overall study design looks good.
Author Response
This manuscript has been polished by professional scientific editing service.
Reviewer 3 Report
The manuscript of Wu et al. well demonstrated the weaknesses of the various methods in the logP calculations and the necessity of the correct choice of the descriptors in QSAR methods. However, it is unclear what was the real aim of this work. Unfortunately, apart from some minor issues, there are weaknesses in the data visualizations which make the reading difficult. Before the resubmission, a native English chemist revision is also necessary.
Some points proposed for reconsideration:
- Please replace "skin permeability" and " quantitative structure-activity relationship (QSAR)" in the Keywords section because the title already contains them, and so there is no added value in this section.
- The authors wrote just in the first sentence of the Introduction, "The skin, which is the largest organ in the human body ..." In which terms are the largest? Surface, weight, etc.?
- The Δ symbol represents a difference rather than a gradient.
- Interestingly, the authors also used software more than 15-years old in their work, but the correlation between the various models and results is unclear between the softwares and the found results.
- The selection rationale of the model compounds is also unclear. There are some chemicals whose presence among the selected molecules is difficult to interpret. An example is diethyl ether. This molecule is highly volatile and has a boiling point close to the natural temperature of human skin, meaning it interacts with the skin only in extreme conditions. Only the minority of the diethyl ether can penetrate through the skin, while its majority immediately evaporates - except, of course, the soaking. Other interesting molecules are the halogenated acetic acids. They are not medicines, but it is also hard to find formulations in which these compounds are present in significant quantities.
- Although the authors defined the r^2 values in the text, the r values are in figures. The referee understands that the r values are larger than the r^2, and so the results are more attractive, but it is incorrect - at least by the referee.
- What do the points (and lines) represent in Figure 4. It is clear that only some data points are in the figure and not the entire set, but the lines and the caption make the figure confusing. What do they represent? Are they a subset of the originals? If so, what were the criteria of selection?
- In Figure 5, two faults make it confusing. The first is that the legend does not have colors, only the circle, square, and triangle. The second is that the squares and triangles have identical colors.
- Figure 6 also contains two errors. The first is in the graph. There is an illegal "@" sign in front of the 0.78, and the second is a typo in the caption.
- The referee would suggest replacing Figure 8 with another, more informative figure. Perhaps more interesting would be the error of the logP calculations versus the measured logP.
- Figure 9 caption writes the Vm vs. 0χ, but the ordinate identifier is log Kp.
- The graphs in the SI file need separate captions because there are so many graphs that the reader can easily get lost among them.
Though the listed concerns do not have equal weights, a major revision is necessary to correct them.
Author Response
See the attached file for detail.

Round 2
Reviewer 3 Report
See the attached file.

Author Response
See the attached file for detail
